# Tuning the Morphology in the Nanoscale of NH_4_CN Polymers Synthesized by Microwave Radiation: A Comparative Study

**DOI:** 10.3390/polym14010057

**Published:** 2021-12-24

**Authors:** Cristina Pérez-Fernández, Pilar Valles, Elena González-Toril, Eva Mateo-Martí, José Luis de la Fuente, Marta Ruiz-Bermejo

**Affiliations:** 1Centro de Astrobiología (INTA-CSIC), Departamento de Evolución Molecular, Ctra. Torrejón-Ajalvir, km 4, Torrejón de Ardoz, 28850 Madrid, Spain; crisperez@cab.inta-csic.es (C.P.-F.); gonzalezte@cab.inta-csic.es (E.G.-T.); mateome@cab.inta-csic.es (E.M.-M.); 2Instituto Nacional de Técnica Aeroespacial “Esteban Terradas” (INTA), Ctra. Torrejón-Ajalvir, km 4, Torrejón de Ardoz, 28850 Madrid, Spain; vallesgp@inta.es (P.V.); fuentegj@inta.es (J.L.d.l.F.)

**Keywords:** HCN polymers, cyanide polymerization, microwave-driven polymerization, nanoparticles, nanofibers, multifunctional materials

## Abstract

A systematic study is presented to explore the NH_4_CN polymerization induced by microwave (MW) radiation, keeping in mind the recent growing interest in these polymers in material science. Thus, a first approach through two series, varying the reaction times and the temperatures between 130 and 205 °C, was conducted. As a relevant outcome, using particular reaction conditions, polymer conversions similar to those obtained by means of conventional thermal methods were achieved, with the advantage of a very significant reduction of the reaction times. The structural properties of the end products were evaluated using compositional data, spectroscopic measurements, simultaneous thermal analysis (STA), X-ray diffraction (XRD), and scanning electron microscopy (SEM). As a result, based on the principal component analysis (PCA) from the main experimental results collected, practically only the crystallographic features and the morphologies in the nanoscale were affected by the MW-driven polymerization conditions with respect to those obtained by classical syntheses. Therefore, MW radiation allows us to tune the morphology, size and shape of the particles from the bidimensional C=N networks which are characteristic of the NH_4_CN polymers by an easy, fast, low-cost and green-solvent production. These new insights make these macromolecular systems attractive for exploration in current soft-matter science.

## 1. Introduction

The development of new smart and multifunctional materials is currently encouraged to be constrained toward the design of low cost and free-solvent synthetic processes, or at least green-solvents, and easy, fast, effective and robust productions. The MW-driven polymerization of NH_4_CN may be considered as a good example of this kind of processes for the generation of polymeric macrostructures with potentially versatile properties. One of the advantages of this polymerization technique is to reduce the reaction times with respect to the conventional heating processes. However, the kinetic variations during the syntheses due to the effect of the MW radiation must be taken in consideration, as they may change the final properties of the products. In the case of the HCN-derived polymers, it is well established that their properties are strongly conditioned by the synthetic conditions and for extension, their potential applications are directly dependent on their chemical composition and structure, thermal stability, particle shape and size, among others.

Recently, it has been shown that the particle morphology of the known HCN-derived polymers can be tuned by the choice of the synthetic conditions [1,2,3]. Some polymers of this heterogeneous family [4] have been proposed as emergent materials with different applications, such as photocatalysts [5], semiconductors, nanowires, ferroelectric materials [6], capacitors [1], coatings with potential biomedical applications [2,7,8,9], protective films against corrosion [10,11], or for the development of antimicrobial media for passive filtration [12].

Different HCN-derived polymers present some structural characteristics resembling those of the extensively researched carbon nitrides [1,13,14], which are used as carbon materials in multiple fields [15,16,17,18]. Moreover, it has been proposed that the incorporation of HCN-derived polymeric nanoparticles as novel fillers in different polymeric matrixes could lead to the creation of new composite materials [1]. However, only one work has described the production of nanoparticles derived from HCN [13]. In that case, the aqueous polymerization assisted by the MW radiation of cyanide lead to the simultaneous generation of nanoparticles with different sizes and shapes, together with isolated long nanofibers. In this context, it is important to note that HCN polymer nanofibers have attracted considerable attention due to their attractive features, e.g., their high visible-light photocatalytic activity. These materials were prepared from formamide, the hydrolysis product of HCN, through conventional thermal heating at 200 °C and a prolonged reaction time of 48 h [5]. Therefore, these polymer materials, at the nanoscale, present an extraordinary potential in the different areas mentioned previously.

Taking into account the high potential of HCN-derived polymers for the development of multifunctional materials, as well as the key role of their polymeric morphology for these potential applications, several aspects related with the MW-driven synthesis of NH_4_CN polymers are explored here in detail, paying special attention to the relationships between the experimental conditions and the features and properties of the final polymeric products. In this way, two series of NH_4_CN polymers synthetized using MW radiation were compared between themselves and also against a control polymer produced under conventional heating using equivalent reaction times or by fixing this time, in order to determinate the influence of the MW radiation on the features of the insoluble polymers obtained.

Moreover, statistical methods were used for a better interpretation of the results reported in the present work. The obtained data were comparatively discussed in order to obtain useful synthetic conditions for the production of cyanide polymers using MW radiation, taking into special consideration the structural characteristic of the polymers and their morphological and textural properties for the further development of multifunctional materials.

## 2. Materials and Methods

### 2.1. Synthesis of the NH_4_CN Polymers

All of the NH_4_CN polymers synthetized in this work were produced as was previously described in [13] using the reaction times and temperatures indicated in Table 1. The initial equimolar concentrations of NaCN and NH_4_Cl always were 1 M. Figure 1 shows some experimental details of the synthesis process.

### 2.2. Characterization of the NH_4_CN Polymers

The Fourier transform infrared (FTIR) spectra were recorded as was reported in [1], but using a Nicolet Is50-FTIR spectrometer (Thermo Scientific, Waltham, MA, USA). The powder X-ray difracction (XRD) was performed as in [1], but by scanning the samples from 10° to 50° (2θ) and with a count time of 3 s. All of the parameters and the equipment for the measurements of the elemental analysis, thermal analysis and ^13^C solid-state cross polarization/magic angle spinning nuclear magnetic resonance (CP/MAS) NMR were described in [1]. The X-ray Photoelectron Spectroscopy (XPS) analysis of the samples (pure pellets of the polymers) was carried out in an ultra-high vacuum chamber equipped with a hemispherical electron analyser, using an Al Kα X-ray source (1486.6 eV) with an aperture of 4 mm × 7 mm (Specs company, Berlin, Germany). The base pressure in the chamber was 5 × 10^−9^ mbar, and the experiments were performed at room temperature. The following core level peaks were recorded under the same experimental conditions: C (1 s), N (1 s) and O (1 s). The pass energy applied to take the overview sample was 30 eV, while 20 eV pass energy was applied for the fine analysis of the core level spectra. The surface morphologies of the polymers were determined by a ThermoScientific Apreo C-LV field emission electron microscope (FE-SEM) equipped with an Aztec Oxford energy-dispersive X-ray microanalysis system (EDX). The samples were coated with 4 nm of chromium using a sputtering Leica EM ACE 600. The images were obtained at 10 kV. This SEM study was analogous to those detailed in [13]. The values of the Z-average and polydispersity index (PdI) were registered using a Zetasizer Nano instrument (Malvern Instruments Ltd., Almelo, Netherlands), using ethanol as a solvent to scatter the samples.

### 2.3. Statistical Analysis

Three principal component analyses (PCAs) using the polymers’ physicochemical characteristics as variables were carried out. The tests were performed using the multivariate data analysis software CANOCO 4.5 (Microcomputer Power, Ithaca, NY, USA) [19]. The program CANODRAW 4.0 (in the Canoco package) was used for the graphical presentation.

## 3. Results and Discussion

### 3.1. MW Radiation Effect in the Production of NH_4_CN Polymers

In order to explore the effect of the MW radiation in the aqueous polymerization of NH_4_CN, two set of experiments were carried out. The first one used equivalent reaction times (polymers **1–5**, Table 1); the second one fixed the reaction time at 67 min (polymers **2**, **6–9**, Table 1). For the first series of syntheses (series 1), the reaction times were chosen with a comparative proposal in a relationship with the NH_4_CN polymer obtained at 80 °C using conventional heating and a reaction time of 144 h (control polymer) [1]. Based on the manufacturer specification of the MW reactor, 6 days under a classical thermal treatment is equivalent to the short reaction times using MW radiation indicated in Table 1 (series 1). On the other hand, the fixed time for the second series (series 2) was based on the fact that, in order to probe a polymerization time shorter than 309 min (polymer **1**) but longer than those explored previously in the NH_4_CN polymerization assisted by MW at 180 °C, with a reaction time of up to 30 min [13]. Thus, the reaction time for polymer **2** was chosen for polymeric series 2.

The conversion degrees, α (%), reached for each polymer are displayed in Figure 2a, and they were calculated as is described in [1] by means of gravimetric measurements of the gel fraction and the insoluble product. The use of conventional heating demonstrated that the increase in the temperature leads to a decrease in the yield of the NH_4_CN polymers [14]. For the control polymer, the conversion was about ≈38%, while this value decreased to ≈17% when the polymerizations are conducted at 180 °C using MW radiation [13]. Note that the higher conversion was found in series 1, around 25%, at the lowest temperature under study 130 °C, and that the data of series 1 is in agreement with these previous results. However, a slight increase in the conversion for the polymerizations was observed at temperatures higher than 180 °C. For series 2, at 130 °C, a decrease of practically five times in the polymerization time produced a relevant and unexpected increase in conversion. On the contrary, an increase in the reaction time at temperatures of 170 and 205 °C causes an increase in the yield of insoluble polymer. The conversion values for polymers **6**, **7** and **9** was really unexpected, reaching yields similar to those obtained by means of a traditional heating process [1,14]. This behavior of the polymerization in the presence of MW radiation seems to indicate significant differences with respect to the kinetic approach reported using conventional heating, in which a clear limit conversion was observed after a concrete reaction time for a fixed temperature [4]. Further works are in progress to study in detail the kinetic behavior of the MW-driven polymerization of cyanide.

In addition, the pathways proposed to explain the formation of NH_4_CN polymers in aqueous media suggest collateral reactions during the course of this precipitation polymerization, such as deaminations, denitrogenations, dehydrocyanations, hydrolysis and oxidations. These last processes are responsible for the incorporation of oxygen in the macrostructures [4], which would lead to a gain of weight considering the total initial mass input in the system (the initial weight of NaCN plus NH_4_Cl). Figure 2b shows the weight balance found in both series, considering the reaction conditions shown in Table 1, based on the following formula: Weight Balance (%) = [(weight NaCN + weight NH_4_Cl) − (gel fraction weight + freeze-dry sol content)]/(weight NaCN + weight NH_4_Cl)]·100. In any case, no direct relationship could be found between the conversion values and the weight balance, as can be observed when comparing Figure 2a,b. In series 1, an increasing weight loss was found as the temperature increased; however, in series 2, when the polymerization time was kept constant, a dissimilar behavior was observed. In this case, at low temperatures, the weight loss, around 5%, was maintained throughout these polymerizations. However, the MW reactions at 170, 190 and 205 °C occurred with substantial weight gain, close to 15% or higher. The increase of weight might be related with a minor loss of nitrogen and/or to a higher increase of mass by oxygenation, but the results of the elemental analysis were inconsistent with this proposal (see below), which suggests that complex secondary processes have no reflex directly in the generation of the gel fraction macrostructures.

The elemental analysis data displayed in Figure 2c indicates that the C percentage did not present significant differences among the two series under study, and it was about 41%. The same behavior was observed for the %O, at around 17%. Only a slight decrease in the nitrogen content was observed in series 2 with the increase of the temperature, from 41 to 37%. This result was also reflexed in the C/H and C/N molar relationship for this series, when the reaction time was fixed, going from 1.18 to 1.31 for this last ratio (Figure 2d). Therefore, it seems that the increase in the temperature favors the deamination processes during the polymerization based on the lower %N and %H. This fact was in agreement with previous results [13]. In contrast, for series 1, no significant variations were found in the molar relationships with the increasing of the polymerization temperature.

As a result, it can be concluded that a temperature of 170 °C seems to be a key parameter in these polymerizations under MW radiation, and also that low polymerization temperatures, such as 130 °C, present a singular behavior. However, the compositional data do not give a clear response to the high weight gain observed in series 2.

At this point, it was necessary to point out that the lower yields for the insoluble polymers may be due to the hydrothermal conditions achieved by MW radiation, which favors the oxidation, hydrolysis and decomposition processes of the NaCN and/or other intermediate reaction products, as in Figure 1, decreasing the availability of them, and therefore reducing the final amount of insoluble polymers collected. Note that all of the reactions were prepared under ambient conditions of pressure and moisture, and neither oxygen or CO_2_ were removed from the reaction vessel or the solvent.

The left part of Figure 1 shows the possible processes which can decrease the available amount of cyanide to polymerize. Atmospheric oxygen can lead to the partial oxidation of cyanide at elevated temperatures, generating cyanate. CO_2_ can react with cyanide to produce carbonate and HCN. In addition, the hydrolysis of the cyanide solutions under heating can produce formiate and ammonium. Under the conditions considered herein, the initial cyanide solutions have a pH of 9.2, and a ~50% of HCN was present in the solutions, which can be hydrolyzed to give formic acid and ammonia. The generation of volatile compounds such as HCOOH, NH_3_ and even HCN may explain the negative weight balances observed in the NH_4_CN polymerization above studied. However, the increase observed for the production of insoluble polymers **6**, **7** and **9** might be due to the recycling of the delivered HCN and NH_3_, generated first as by-products, that finally were implicated in new processes of oligomerization/polymerization, as a closed system was used in the described reactions. On the other hand, the positive weight balances can be due to the formation of no volatile oxygenated molecules such as carbonate and cyanate, as this takes place at high temperatures within series 2. Besides this, in order to explore whether the ambient conditions have a significant influence in these plausible side reactions, where the atmospheric oxygen and the CO_2_ were implicated, two additional reactions were carried out. Polymers **3** and **7** were again synthetized, but using an inert atmosphere of nitrogen and carefully removing the CO_2_ and O_2_ of the water by an N_2_ stream. The following conversion values were obtained (syntheses were made twice) α(%) = 7.7 ± 0.1 and α(%) = 3.8 ± 0.3 for the analogous polymers **3** and **7**, respectively. Furthermore, in both cases a loss of mass was observed in the total weight balance. Note that the effect of the absence of air in the conversion obtained was outstandingly significant for the analogous polymer **7**. Therefore, the lack of air in the reaction environment does not seem to prevent the side reactions in the cyanide polymerization assisted by MW radiation; on the contrary, the yield of the reaction was remarkably lower. Due to the great impact of the air in the microwave-cyanide polymerizations, new works will be developed to obtain a full understanding of this factor in these highly complex polymerization reactions.

In addition, we might take into consideration that other hydrolysis and/or oxidations of some the intermediate products during the polymerization process finally lead to the lack of formation of the insoluble polymers, as is the case for diaminomaleonitrile (DAMN). DAMN, the tetramer of the HCN, was considered to be a main intermediate in the generation of the HCN polymers (Figure 1) (see, e.g., [4]). However, this compound can be oxidized to diiminosuccinonitrile in water, and this is finally decomposed into oxalic acid and urea (upper part of the Figure 1). Furthermore, DAMN in an aqueous medium can be transformed into formamide, glycine and aminomanolic acid (the bottom part the Figure 1), as was revealed by Ferris and Edelson [20].

The higher yields (~75%) in the hydrothermal polymerizations of DAMN than were the obtained in analogous NH_4_CN polymerizations (35–40%) using conventional heating for the production of black insoluble HCN-derived polymers [1], and the significant diminution of these yields in the MW-driven polymerization of DAMN (~30%) (a further work is in progress; no data are shown here) allows us to reaffirm the following points: (i) the DAMN was a main intermediate product cyanide polymerization, as similar—although not identical—final products can be obtained in both ways, i.e., the cyanide and DAMN polymerizations produced resemble—but are not identical—to HCN-derived polymers [1]. The possible polymerization of cyanide by other different DAMN pathways through its dimer and/or its trimer, aminomalononitrile (AMN), for the generation of the suggested C=N networks of the HCN-derived polymers must also be taken in consideration (Figure 2); (ii) the higher yields obtained for the DAMN polymers seem to indicate that the cyanide undergoes several processes in water, which prevent the generation of DAMN and the subsequent polymerization of this oligomer. Therefore, it seems that the cyanide polymerization via the DAMN pathway was the main process.

As a summary of the above, one can say that: (i) the lack of air in the cyanide polymerization assisted by MW radiation does not seem to inhibit the side reactions which prevent the generation of DAMN, and (ii) the decomposition processes of the DAMN are related to the presence of water. The increase of the temperate increases the ratio of DAMN decomposition in water environments. Both aspects are the main focus of other works in progress.

The FTIR spectra of both series closely resembled those reported from the NH_4_CN polymerizations under a classical heating and also assisted by MW at 180 °C (Figure 3a,b) [1,4,13,14]. In these previous works, analyses of the features for these spectra were discussed in detail. The greatest difference between the spectral features of series 1 and 2 was the relative intensity of the bands related with the nitrile groups (-C≡N), around 2200 cm^−1^. This difference was clearly observed when the EOR values (the extension of the reaction, EOR = [I_1645_/(I_1645_ + I_2200_)] 100) [22] were represented (Figure 3c). For series 1, this value decreased with the temperature, and by the contrary, for series 2 the behavior was the opposite. This quantitative spectroscopic parameter was directly related to the progress of the polymerization reaction under traditional heating conditions, i.e., the increase of the monomer conversion along the reaction time gives noticeably higher EOR values [22]. However, there was no observed clear relationship between the EOR and the conversion degree, especially in series 2 (Figure 3d, upper part). This behavior was also observed from the NH_4_CN polymers synthetized at 180 °C by MW [13].

**Scheme 2 polymers-14-00057-sch002:**
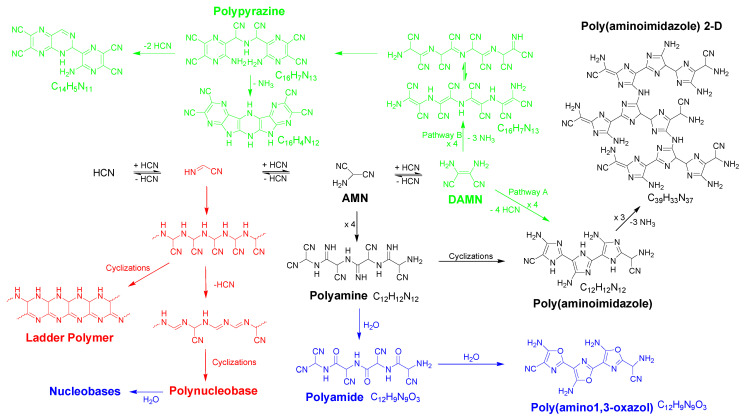
General pathways for the generation of HCN-derived polymers in water. In this scheme is shown the oligomer formation as far as the tetramer DAMN, as well as proposed macromolecular structures from the dimer (red color), the trimer (AMN) (black color) and DAMN (green color) according to recent works [1,3,4,23]. Note that the incorporation of oxygen in some of the structures for the suggested macromolecular systems was due to hydrolytic processes (blue color).

In addition, the calculation of the EOC (extension of the conjugation, EOC = [I_1645_/(I_1645_ + I_3330_)]·100) [1,13] indicated no significant differences between series 1 and 2, with the exception of the values determined at 170 °C (Figure 3c). Likewise, there was no clear trend found when these values were confronted with the conversion. Again, these spectroscopic data revealed that 170 °C shows a remarkable singularity with respect to the rest of the temperatures under study.

Simultaneous thermal analysis (STA) for representative samples of polymers **1****–9** was carried out (Figure 4), and some parameters obtained from these measurements are listed in Table 1. All of the samples showed the same amount of moisture, at about 10%; however, while the NH_4_CN polymers from series 1 presented a similar content of char residue, 19–22%, the samples from series 2 exhibited an increase of the char from 13 to 25% with the increase of the temperature. Polymer **9** was the sample that presented the highest percentage of char, and by the contrary, polymer **6** registered the lowest value. In general, a greater amount of char was related with highly cross-linking macrostructures and rigid chains; in particular, NH_4_CN polymers were associated with the presence of cross-linking oxygenated groups, such as inter- and intramolecular amide bonds [4]. The C/H, C/N, C/O and N/O molar ratios were 0.9, 1.2, 4.1 and 3.5, respectively, for polymer **6**; and 1.0, 1.3, 3.5 and 2.7, respectively, for polymer **9**. Thus, it seems that the higher thermal stability of sample **9** was related with greater oxygenated macrostructures, and with a minor amount of nitrogenized groups, but not with a more conjugated structure, considering the C/H values and the EOC data. Therefore, it might be proposed that this stability may be due to the presence of oxygenated cross-linking groups.

On the other hand, the DTG curves showed similar profiles to other NH_4_CN polymers synthetized at 80 °C [1]. Five characteristic peaks were observed: (i) at ~70 °C (water desorption); (ii) at 220–260 °C and at ~400 °C (thermal break of the weakest bonds); and (iii) at ~670–680 °C and 820–840 °C (high thermal decomposition, carbonization stage). Only a slight difference was found for polymer **5** with an additional degradation peak at 477 °C, and the decrease of the relative peak intensity at ~260 °C in series 2 with the increase of the temperature. In addition, the DSC curves were really dissimilar to those registered for NH_4_CN polymers synthetized at lower temperatures [1,4], proving that the thermal analysis techniques provide excellent fingerprints to distinguish NH_4_CN polymers with very similar FTIR spectra [4]. For both series of experiments, the first endothermic peak, at 75–85 °C, can be related to the loss of the absorbed water in the polymeric matrix, and the rest of peaks can be related to degradative processes.

Some morphological aspects, such as the hydrodynamic diameter, were measured for representative polymers **1–9** (Z-average, Table 1). Considering this parameter, for series 1, the particle size was decreased with the increase of the temperature, reaching nanoparticles with a diameter of ~280 nm and with relatively low PdI = 0.22–0.25. On the contrary, in series 2, although an apparent decrease in the molecular size may be related with the increase of the temperature, the PdI values were higher than those in series 1, indicating a higher degree of heterogeneity. Indeed, the data from sample **9** were not properly registered due to a presumable aggregation of the particles.

The XRD pattern profiles for NH_4_CN polymers **1–9** have the same look as those previously reported for analogous polymers synthetized using a conventional hydrothermal procedure or MW radiation [13,14] (Figure 5a,b). It was expected that an increase in the temperature would lead to a more ordered macrostructure [13,14]. This is the unique observed peak (2θ ~27°) related to graphitic-like two-dimensional (2-D) structures, such as layered g-C_3_N_4_ [16,24], would be higher and narrower at the higher polymerization temperatures. However, this prediction was not valid for the range of temperatures under study when analyzing the data of the crystallinity of the different samples collected in Table 1. There was no clear relationship between the order level of the macrostructures with the increase of the temperature or the reaction time. All the samples exhibited an average crystallinity values of 65%, with the exception of polymers prepared at 130 °C, with values higher and lower values than this; showing a particular behavior of the cyanide polymerization at this temperature at least related with the internal order of the macrostructure.

It was clearly shown above that the heating by MW radiation has an unexpected and significant influence in the behavior of the cyanide polymerization, providing results which were not previously observed with conventional heating, i.e., there was not a proportional direct relationship between the reaction time and the conversion degree (%) and EOC [14,22], or temperature and crystallinity [14], for example. However, the final reaction products present resemble the spectroscopic and thermal characteristics of analogous HCN-derived polymers synthetized using conventional heating; for the size of the particles and the higher order of the macrostructures, this was the main difference found between the conventional heating against MW radiation. No nanoparticles were identified in NH_4_CN polymers synthetized using conventional heating (the Z-average and PdI could not be measured; the data are not shown; however, see [1] for SEM images); at the lower temperature herein considered, the greater size of particles were observed. In addition, at 130 °C and 170 °C, particular features were observed with respect to the syntheses carried out at the other temperatures. The global and proper interpretation of all of these results is not simple and trivial. Thus, multivariate analysis could be an excellent analytical tool to reach a better and more objective interpretation of the data presented above, and could help us to choose preferential conditions to tune the nanoscale of the NH_4_CN polymeric particles. In this way, a PCA was carried out to reach a first full overview of the MW radiation effect in the NH_4_CN polymerization behavior (Figure 6). For this analysis, the total data for the nine polymers (Table 1) were considered. The conversion degree, the balance weight, and all of the molar ratios were calculated; the EOR, the EOC, the crystallinity, the char content, the moisture percentage, the Z-average and the PdI were taken in consideration, explaining 99% of the variance. The first axis shows the highest positive correlation among a wide number of variables, with the exception of EOC and the balance weight. The right-hand portion of the first axis was predominantly occupied by EOR, moisture, monomer conversion, C/O molar ratio, Pdl, Z-average and N/O molar ratio; the left-hand portion was occupied by crystallinity (%) and the C/N molar ratio. The second axis indicates the highest positive correlation among char (%) and the C/H molar ratio. By contrast, the EOC and balance weight exhibited the highest negative correlation with this axis. From these statistical results, one can say that the there is a direct relationship between the conversion degree and the EOR values, as was expected, although a clear relationship was not observed in Figure 3d. In addition, a higher conversion degree was directly related to less-oxidized structures; interestingly, this minor oxidation degree was directly related with a higher size of the particle and a higher polydispersity degree. On the other hand, the percentage of char (%) was strongly correlated with the molar C/H ratio, indicating that the more conjugated structures are the more thermally stable, but apparently independent of the %O in the structure. A higher internal order based on the percentage of crystallinity is directly related with the C/N ratio and with the char (%), i.e., the more carbonaceous and more conjugated structures are the more crystalline and also the more thermally stable, but the Z-average and crystallinity (%) are independent variables. Therefore, if the MW radiation is considered to obtain nanoparticles of NH_4_CN polymers, as under conventional heating these are not formed and MW notably reduces the reaction times, it seems that reaction conditions which lead preferentially to the oxidization of structures but are highly conjugated should be considered.

With respect to the polymeric samples, PCA analysis indicated groupings of the NH_4_CN polymers into well-differentiated sets: (i) one corresponding to series 1 (polymers **2**, **3**, **4** and **5**), with the exception of polymer **1**; (ii) another one with polymer **7** and **8**; (iii) and polymers **1**, **6** and **9** isolated from the rest, indicating a particular behavior, as was expected especially for **1** and **6**, as they were the polymers synthetized at 130 °C, although polymer **1** was next to the group of series 1, in agreement with a relative resemblance with the samples of this group. In relation to the first group, the clear grouping of polymers **2**, **3**, **4** and **5** indicated that, statistically, the choice of the equivalent reaction times for different temperatures according to the instruction of the manufacturer of the MW reactor, and considering the experimental results indicated above, gives NH_4_CN polymers with similar characteristics. The second group, formed by polymers **7** and **8,** showed higher char (%), C/H and C/N molar ratios and crystallinity (%), indicating that these polymers are thermally stables carbonaceous structures. For the first, the polymers **6** and **9,** and to a lesser extent **1**, showed a higher EOR, moisture (%), α (%), C/O molar ratio, Pdl, Z-average and N/O molar, i.e., they presented the polymeric particles with the highest size and lowest oxygen content. Therefore, the lowest and the highest temperature here studied do not seem adequate for the development of nanoparticles.

These well-defined groupings by similar and resemblance features revealed by the PCA analysis, in a certain way, could also be observed for the shape differences found by SEM (Figure 7). Thus, polymers **2**, **3**, **4** and **5** presented rice-shaped nanoparticles and other stacking oval particles together with isolated nanofibers. Polymers **7** and **8** showed clear groupings of nanofibers together with other particles with other shapes (please see Appendix A for details), whereas polymer **9** was estranged from this set in the PCA analysis, showing isolated nanofibers and undefined-shape particles. Polymers **1** and **6**, synthetized at the lowest temperature, presented a particular behavior. Polymer **1** displayed similar rice-shaped nanoparticles and staking oval particles to the rest of the polymers from series 1, but also spherical particles similar to those identified in the control polymer [1]. The particles observed in polymer **6** were similar to the particles found for series 1, although the staking oval particles were not identified. These SEM images also seemed to indicate that 130 °C and 205 °C are not optimal temperatures to produce nanoparticles from cyanide, which is in agreement with the PCA results.

### 3.2. Structural Comparative Study between NH_4_CN Polymers

In order to complete this study about the effect of MW radiation on cyanide polymerization, a detailed structural comparison between the control polymer and polymer **3**, as a representative sample of series 1, was carried out. The previous results for this series indicated that polymers **2**, **3**, **4** and **5** are very similar, as was expected considering the equivalence of the reaction times; they resembled polymer **1,** but were at least morphologically very different to the control polymer. Thus, this section can help us to obtain a comprehensive knowledge about the MW heating role in the polymerization of the NH_4_CN when equivalent polymerization times are considered on the spectroscopic and thermal properties, as the morphological differences clearly showed above.

The comparison of the data from the control polymer and from polymer **3** indicated that the conversion degree decreases notably when the NH_4_CN polymerization is assisted by MW radiation, as it was explained above, but no significant elemental composition variations were observed. For the control polymer, the elemental compositional data were %C 41.4 ± 0.3, %H 3.8 ± 0.2, %N 40.1 ± 0.4 and %O 14.7 ± 0.7, and for polymer **3** they were %C 40.3 ± 0.6, %H 3.5 ± 0.1, %N 39.3 ± 0.8, %O 16.7 ± 1.5 (taking into consideration at least three samples synthetized independently). The subtraction of the normalized FTIR spectra of both samples does not indicate significant differences among them, except for a few low-intensity features (Figure 8a). Some of these bands can be related with the resonances found when the corresponding ^13^C NMR spectra were subtracted (Figure 8b). The FTIR band centered at 2163 cm^−1^ can be related with the resonance at 115 ppm assigned to nitrile groups, the band at 1720 cm^−1^ with the resonance at 151 ppm associated with carbonyl groups, and the bands at 3615 and 3495 cm^−1^ with the signal at 51 ppm related to hydroxyl groups. However, the relative intensity of these FTIR bands and resonances seems to point to there being no great differences between the two polymers.

The XRD analyses showed the same diffraction, but in the case of the polymer **3**, this peak was higher and narrower, indicating a more ordered structure (Figure 8c). In addition, the analysis of the second derivative of their corresponding TG curves showed a very resemble thermal behavior (Figure 8d) which would indicate similar macrostructures, as both polymers present the same thermal behavior. Only the decomposition step at 278 °C may be more noticeable for the polymer **3**. This thermal decomposition step would be related, based on the TG-MS curves (data no shown), with the fragment *m*/*z* 44 which can be assigned to the loss of CO_2_ and/or HC(=NH)NH_2_ or HCONH- (a detailed discussion of the TG-MS results is out of the scope of the present work, and it will be given in a further paper). The slightly higher delivery of CO_2_ or HCONH- for polymer **3** is in good agreement with the spectroscopic data and elemental compositional data indicated above, indicating a higher content of oxygen in polymer **3**.

In the light of these results, the control polymer and the polymer **3** seem to resemble one another, except for very little differences related with the amount of oxygenated functional groups. Thus, detailed XPS analyses of these two samples were made in order to provide further information about them. Figure 9 shows the core-level spectra of the C (1s), N (1s) and O (1s) peaks, as well as their deconvolutions on different components of the control polymer and polymer **3** samples.

A deconvolution study of the C (1 s) peak showed three components for both cases: the first component at 285.1 eV (binding energy) was attributed to the C adventitious, C–H and C–C group; the second component at 286.7 eV corresponded to C-N, C-O, C=N and amide groups; whereas the third component was observed at 288.5 eV, and was assigned to the C=O and nitriles groups. Both samples showed similar carbon components, and the ratio between the components was also comparable; polymer **3** showed a slight increase of 10% for the first component and a decrease of 10% for the second. Thus, the resemblance between spectra C (1 s) seems to indicate very similar macromolecular structures for polymer **3** and the control polymer. The N(1 s) peaks of both samples were resolved into two components, the first one at 398.9 eV being assigned to -CONH_2_ and imines (-N=C<), which were predominant in the control polymer sample, and the second one at 400.0 eV corresponding to -CONH- groups, amides and nitriles; both nitrogen components showed a similar percentage for polymer **3’s** case. Regarding the O (1 s) peak, we fitted the experimental data points using three components. The first component occurred at 530.9 eV, which was possibly assigned to the carboxylate group and to the amide group (-CONH-), which was predominant in the control polymer’s case; the second one appeared at 532.0 eV, but it was mainly a contribution from contamination during the sample preparation in air instead of under UHV conditions; the main component for the polymer **3** sample, and finally a third component at 533.7 eV assigned to C=O and COOH groups, were similar in both cases. Therefore, this comparison study did not show large differences between both samples. The overlapping of several functional groups at similar binding energies did not help us to make an unambiguous assignment for the complex functional group mixture present in the NH_4_CN synthetized polymer’s structure. Nevertheless, the carbon and oxygen components related to the adventitious are more intense for polymer **3**, whereas for the control polymer, the carboxylate and C=N functional groups seems to be the principal component of the analysis.

As a result, taking into account the comparative results between the control polymer and polymer **3**, the more significant effects of the NH_4_CN polymerization assisted by MW radiation are the decrease on the conversion degree and variations in the textural and morphological properties of the final products (as was indicated above, please see the SI of [1] for the look of the control polymer particles; these ones were not nanoparticles). The MW radiation leads to the generation of nanoparticles and/or nanofibers of cyanide polymers in minor yields, but with similar compositional/structural characteristics and the same thermal properties with respect to those microparticles produced under conventional heating conditions and with a more ordered macrostructure. Thus, nanoparticles/nanofibers can be obtained using MW radiation by aqueous cyanide polymerization. The size, shape and polydispersity of these particles seems to be directly related with the reaction time and with the temperature. In order to deepen this result, in the next section, analyses of the morphology of a series of NH_4_CN polymers synthetized at 170 °C and distinct reaction times are given.

### 3.3. Polymeric Particles’ Morphology Variations along the Reaction Time

A relative study of samples synthetized at 170 °C was completed by SEM using different reaction times, from 5 to 120 min. Representative images of these new polymeric series are shown in Figure 10 (for more details, please see Appendix A). We focused on this temperature based on the data reported in the first part of this work, as greater yields were obtained at this temperature; 130 °C and 205 °C were ruled out to explore the production of nanoparticles/nanofibers based on the PCA results, and also due to the easier dispersion of the polymer synthetized at 170 °C in EtOH to prepare the samples for the SEM measurements compared to those synthetized at higher temperatures, i.e., 190 °C and 205 °C. Note that the values of the PdI and Z-average increase with the increase of the temperature and with the reaction time (Table 1), leading finally to molecular aggregates.

All of the samples from the 170 °C series present isolated long nanofibers, with the exception of polymer **7** (reaction time ~ 67 min), which showed a clear grouping of long nanofibers (Figure 7 and Appendix A). Other shapes observed were spherical/oval particles, irregular and planar stacking, rice-shaped nanoparticles and short nanofiber networks. In all of the samples studied, on general lines, there was a dominant morphology against others depending on the reaction time, as is qualitatively summarized in Table 2. Interestingly, it is the generation of short nanofibers at 36 and 52 min which was not observed previously for this type of polymer. On the other hand, the production of rice-shaped nanoparticles was specially improved at 105 min, and was practically the only shape observed. Thus, to highlight, short nanofibers, long nanofibers and nanoparticles were obtained at 170 °C with reaction times of 36, 67 and 105 min, respectively. Therefore, it seems that there is a clear effect of the reaction time on the shape and size of the NH_4_CN polymers synthetized in the presence of MW radiation.

Based on the PCA results discussed above, the more oxidized and conjugated macrostructures derived from the MW-driven cyanide polymerization would be nano-sized. This statistical result could be experimentally proven. Thus, Figure 11 shows the evolution in the composition along the time of the cyanide polymers synthetized at 170 °C. It can clearly be seen that the two samples with a greater content in oxygen, in this case those obtained at 36 and 52 min, present short nanofiber networks, with these being apparently the smallest nanoparticles observed. It is also interesting that the sample obtained at 105 min, with the highest content in oxygen, showed the tiniest particles observed in all of the cases here studied. On the contrary, the sample prepared using a reaction time of 67 min, polymer **7**, presented the lower content in oxygen, and in this case long nanofiber networks were found. Therefore, taking into account these results, it seems possible to tune the morphologies of the NH_4_CN polymers to obtain mainly short or long nanofibers or nanoparticles for the development of new families of polymeric materials. However, the lack of a direct relationship between the conversion degree, elemental composition and reaction time, as was observed in Figure 11, encourages us to carry out comprehensive studies about the cyanide polymerization promoted by MW radiation.

## 4. Conclusions and Outlooks

This is the first systematic study regarding the MW-assisted polymerization of cyanide, addressing a wide range of temperatures and reaction times. As the main results, the MW radiation has a notable influence on the yields of the insoluble polymers obtained. This fact could be due to the likely increase of the decomposition processes for the cyanide, and of its main oligomer, the DAMN, hindering the polymerization pathways proposed for the production of the extended C=N polymeric networks, as postulated for the HCN-derived polymers. However, using equivalent reaction times, polymers with very similar compositional, spectroscopic and thermal properties were obtained. In addition, note that for particular reaction conditions at higher temperatures, the conversions achieved were similar to those using traditional heating systems. Moreover, it is highly informative that the MW radiation allows the generation of HCN-derived polymeric particles at the nanoscale which were not observed previously under any experimental condition using conventional methods; this technique could be successful for the tuning of their morphological properties, and for extension to obtain a new promising family of nanomaterials, taking into account the recent potential revealed by these polymeric systems. Due to the unexpected behavior of the cyanide MW polymerization, the multivariate analysis has turned out to be a successful tool to obtain a global interpretation of the results obtained herein and beyond to help us to find appropriate reaction conditions which lead to the generation of materials with concrete structural and morphological properties.

Being a very fast, robust, easy, low cost and green-solvent process, and the possibility to tune properly the properties of the final products makes the aqueous microwave-driven cyanide polymerization a highly attractive and promising methodology for the generation of new multifunctional materials. However, due to the apparently random behavior of the system along the reaction time, mainly due to the experimental variables to tune the properties of the polymers, it is necessary to carry out exhaustive synthetic and structural studies, and to examine the results under the light of statistical approaches to develop properly cyanide-based materials.

## Data Availability

Not applicable.

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
