# Peer review of "Tuning the Morphology in the Nanoscale of NH4CN Polymers Synthesized by Microwave Radiation: A Comparative Study"

_polymers, 2021, doi:10.3390/polym14010057_

Round 1
Reviewer 1 Report
Cristina Pérez-Fernández et al. reported Tuning the morphology in the nanoscale of NH4CN polymers synthesized by microwave radiation: A comparative study. The work has revealed the NH4CN polymerization under microwave radiation at different temperatures. The topic is highly exciting and the study is very informative. Most of the time microwave radiation helps to improve the reaction condition for small molecules such as high yield, short reaction time, reaction proceed at lower temperature etc. The authors have worked previously on NH4CN polymerization under microwave radiation at a particular temperature (Sci. Rep. 2020). I feel this work is much similar to the (Sci. Rep. 2020). The topic is highly exciting and the study is very informative.
However, the clear advantage of microwave radiation is missing and it requires major revisions in order to meet the journal's requirements.
- The authors have talked several times about kinetics (the kinetics data is missing). It would be nice to see the kinetics data. Kinetics data can provide precise conclusions. The results are not sufficient to provide a systematic conclusion without it. The kinetics data will help the readers a lot.
- CHN analysis brings more ambiguity to the results. I believe more controlled environment polymerization (N2 and Ar environment) would be interesting. These reaction environments will reduce the side reactions and oxidation. The controlled experiments will provide enough data for the advantage of microwave radiation.
- The authors can try to perform microwave polymerization at different pressure.
- The UV-vis Spectroscopy can provide more insights into the polymers.
- 13C NMR for all the polymers will help to characterize and support the claim.
- Few spelling mistakes table 1. Crystallinity not cristallinity.
- DOI links are missing in the references.
Author Response
Thank you very much for these helpful points raised by the reviewer. We have changed the manuscript according to some his/her suggestions, point by point (we have highlighted the changes in blue in the new version).
Cristina Pérez-Fernández et al. reported Tuning the morphology in the nanoscale of NH4CN polymers synthesized by microwave radiation: A comparative study. The work has revealed the NH4CN polymerization under microwave radiation at different temperatures. The topic is highly exciting and the study is very informative. Most of the time microwave radiation helps to improve the reaction condition for small molecules such as high yield, short reaction time, reaction proceed at lower temperature etc. The authors have worked previously on NH4CN polymerization under microwave radiation at a particular temperature (Sci. Rep. 2020). I feel this work is much similar to the (Sci. Rep. 2020). The topic is highly exciting and the study is very informative.
However, the clear advantage of microwave radiation is missing and it requires major revisions in order to meet the journal's requirements.
The authors have talked several times about kinetics (the kinetics data is missing). It would be nice to see the kinetics data. Kinetics data can provide precise conclusions. The results are not sufficient to provide a systematic conclusion without it. The kinetics data will help the readers a lot.
We agree with the referee’s comment, as strictly speaking the rates of processes is a research area of kinetic. Kinetic research ordinarily pursues two objectives. One objective is to parameterize the rate as a function of temperature and other variables about kinetic analysis. And a second objective is to gain mechanistic insights into the polymerization processes.
In a previous study [old reference 21], a Kamal autocatalysis kinetic model was proposed to describe the formation of insoluble NH4CN polymers and gel fractions during the polymerization of NH4CN in aqueous medium at relatively high temperatures, 75-90 °C and using an conventional heating system. Under those conditions the maximum reaction rate occurred at low conversions, and an autocatalytic kinetic model for these precipitation polymerizations must be considered. According to the behaviour of an autocatalytic polymerization, the equation dα/dt = k αm (1-α)n, where m and n are reaction orders and k is the reaction rate constant, could be used. However, due to the very complex pathway proposed for the cyanide polymerization, where several events occur simultaneously, kinetic models with multiple rate can provide more accurate results. Therefore, the well-known Kamal-Sourour kinetic model was chosen. This model involves two rate constants, k1 and k2 (with two different activation energies and pre-exponential factors), and has the following form, dα/dt = (k1 + k2 αm)·(1-α)n, with n and m being also reaction orders. In this case, the influence of the reaction products on the reaction rate is given by the term k2αm. The application of this phenomenological kinetic model instead of mechanistic models was based on two relevant aspects: on the one hand, the simplicity of the phenomenological models; and on the other hand, as it was indicated above, the extraordinary complexity of the aqueous heterogeneous NH4CN polymerization, which is convoluted with hydrolysis, oxidation and other thermally initiated chemical reactions.
However, certain limitations were found when this phenomenological approach was employed. Low correlation coefficients were obtained, especially at lower polymerization temperatures, and unrealistic reaction orders were determined. In addition, an analysis of the activation energy (Ea) was only performed from three temperatures in a narrow range, and then, a reliable determination of the Arrhenius parameters was not achieved. These results indicated a change in the model with the polymerization temperature and/or with the conversion range under study. This hypothesis resulted in a second contribution where an empirical model had to be used [14].
In the present work only conversion data have been provided at a certain reaction time, which in some way can be considered as kinetic data (Figure 2a and Table 1). The nature of the microwave technique renders it impossible to acquire conversion values for reaction times less than 2 min due to the necessary thermal profile used to reach the desired temperatures. Thus, the gravimetric methodology described in the reference [14] and old reference [21] was totally discarded in the present work, since we cannot be able to follow the system at very short reaction times. Therefore, you can see that the Figure 11a shows the variation of the conversion degree as a function of polymerization time, but we could not calculated kinetic parameters due to the conversion vs time profile. This one could not be fitted to any known kinetic model.
Therefore, in agreement with the reviewer, the term “kinetic” has been replaced throughout the entire manuscript when it was not properly used. A further work is in progress to understand the behaviour of the cyanide polymerization along the time using multivariate analysis since the classical kinetic methodologies do not seem suitable to explain the complexity of the microwave-driven cyanide polymerization.
CHN analysis brings more ambiguity to the results. I believe more controlled environment polymerization (N2 and Ar environment) would be interesting. These reaction environments will reduce the side reactions and oxidation. The controlled experiments will provide enough data for the advantage of microwave radiation.
The % O in the HCN polymers is independent on the atmosphere, air or nitrogen, used during the polymerization processes since the presence of oxygen in the macrostructure is caused by the water no by the presence of atmospheric air (J. An. Appl. Pyr. 2017, 124, 103-112 and Orig. Life Evol. Biosph. 1987, 17, 283–293).
On the other hand, our main hypothesis is that the DAMN compound is the central intermediate in the generation of the HCN polymers, as it is shown in the Scheme 2 of the main manuscript and as it is discussed in the text. Currently, we are studying the DAMN polymerization both, in bulk as an in aqueous suspension using microwave reaction. As preliminary results of both studies, we can say that the presence of air has not a significant influence in the DAMN polymerization. In the particular case of the microwave-driven DAMN polymerization, the air has only influence in the crystallinity of the final products, but not in the yield of the reactions neither in the chemical compositions nor in the spectroscopic and thermal properties of the final products. Therefore, it seems that the absence of air during the DAMN polymerization assisted by microwave radiation does not prevent side reactions. The microwave radiation seems to favour the oxidation and decomposition processes of the DAMN in water (right part of the Scheme 1). Moreover, in a further work also in progress, conversions of DAMN above 80% are reached using other solvents different than water in activated thermal polymerization. These reactions are carrying out in the presence of air. Thus, in this way the side reactions and the oxidation and decomposition processes during the DAMN polymerization seems to be more related with the role of the water than with the presence or lack of air.
However, following the reviewer´s comment, new experiments were carried out. Thus, new data are present about the polymerization of NH4CN under the analogous conditions to the polymers 3 and 7, but using a nitrogen atmosphere and carefully removing the CO2 and the O2 of the cyanide solutions by a stream of nitrogen, previously to the heating in the microwave reactor. Both reactions were carried out at 170 ºC. We have chosen this temperature for these additional experiments, since it was the better temperature found in the present study to tune the morphology of the cyanide polymers. In both cases, the conversions reached using an inert atmosphere of nitrogen were lower than those obtained in the presence of air, a (%) = 7.7 ± 0.1 and a (%) = 3.8 ± 0.3 for the analogous polymers 3 and 7, respectively. Also in both cases, a loss of mass is observed. It is notably significant the effect of the lack of air in the conversion obtained for the analogous polymer 7. Therefore, the absence of air in the reaction environment does not seem to prevent the side reactions in the cyanide polymerization assisted by microwave radiation, by the contrary the yield of the reaction is remarkably lower. Moreover, in general, the presence of air increases the conversion values for the insoluble cyanide polymers (Chem. Biodivers. 2017, 14, e1600241 and Chem. Eur. J. 2016, 22, 12785-12799). Due to the great impact of the air in the microwave-cyanide polymerization future works will be developed to have a fully understanding of this factor in these highly complex polymerization reactions.
As a summary of the above discussed, we can say that: i) the lack of air in the cyanide polymerization assisted by microwave radiation does not seem to inhibit the side reactions, which prevent the generation of DAMN; ii) and the decomposition processes of the DAMN are related with the presence of water. The increasing of the temperature increases the ratio of DAMN decomposition in water environments.
Therefore two paragraphs explaining this point raised by the reviewer has been added in the new version of the manuscript at section 3.1 MW radiation effect in the production of NH4CN polymers (page 6 and page 7).
The authors can try to perform microwave polymerization at different pressure.
We agree with the reviewer that it is an interesting topic and must be studied. However, our microwave reactor only allows us to fix the temperature and the pressure is self-regulated by the reactor based on the temperature chosen, the solvent and the concentration and nature of the reactants. In fact, in Table 1, you can see that the pressure value is different depending on the temperature used in each case. With our microwave reactor for synthesis, only we can chose the temperature parameter, and no the working pressure value. For these kind of experiments we would need another kind of reactor, which is not available in our laboratory.
The UV-vis Spectroscopy can provide more insights into the polymers.
The UV-vis spectra of the polymers 1-9 have been registered together with the polymers from the new reactions carried out under a nitrogen atmosphere and with the control polymer synthetized at 80 ºC. All these spectra are herein showed. All of them present a broad absorption around 375 nm. Nevertheless, we prefer do not take into account these spectra to be discussed in the main manuscript as not significant differences have been found between them. On the other hand, the UV-vis spectra were recorded in DMSO. These cyanide polymers present a very scarce solubility or null solubility in the common organic solvents, then to our knowledge the DMSO is the better solvent for these type of polymeric systems. However, the DMSO is not the best solvent to register UV-vis spectra due to its cut-off. Thus, the registered spectra do not have a good resolution due to the poor solubility of the polymers and to the nature of the solvent. As it is the case, for this second reason, we prefer no include these spectra in the present work.
We think that the only interesting difference observed is that the band centred around 375 nm is broader in the solids obtained under inert atmosphere than under air conditions. However, as it was commented in the previous point, understanding of air effect in these type of reactions deserve a detailed additional study for a complete new manuscript.
Figure shows the UV-vis spectra of the polymers 1-9, polymers 3 and 7 under air and nitrogen atmospheres conditions.
13C NMR for all the polymers will help to characterize and support the claim.
As you can see in the previous version of the manuscript figure 7b, new version 8b, both spectra presented the same broad features, then new spectra will not provide any further information about these complex polymers, as we have seen in previous works (ref. [1] and [4] and Eur. Polym. J. 2022, 162, 11087). An explanation of this finding is that a repeated 2-D structural unit probably exists along, with significant covalent cross-linking, which constrains motions and introduces structural heterogeneity. We think that an expanded study of the 13C NMR spectra will not give any evidence about the modulation at the nanoscale of the cyanide polymers neither about their structure or hypothetical formation pathway. This fact is reaffirmed in the Figure 8b, where you can see that the differences between NMR spectra is really very low, about 5%; and also in the exhaustive FTIR analysis for all the polymers, as it is illustrated in the previous version figures Figure 2a-b and Figure 7a, new version of the manuscript Figures 3a-3b and Figure 8a. In addition, this detailed NMR spectroscopic study is out the scope of the present work, and we think that it will not shed light about the actual nature of the cyanide polymers herein described, beyond of the already discussed along the text..
Few spelling mistakes table 1. Crystallinity not cristallinity.
This mistake have been corrected.
DOI links are missing in the references.
The DOI links have been added.

Reviewer 2 Report
The author present a manuscript regardin the possibility of tuning the morphology in the nanoscale of NH4CN polymers, using MW radiation.
The paper presents all the section completed, the introduction and the conclusion are rich of details and the experimental part is covering well all the aspects. The discussion is well done, all the data presented are commented and overall is well done.
I have only a major concern: on a total of 28 reference 11 cite the corresponding author Ruiz-Bermejo, even if does not exist a maximum I think more than 5 is not correct. Please as in the review in citation 4 you have already cited [21] as 137; [25] as 139; [26] as 140 etc please cite the same review and do not abuse of self-citations.
others minor comments:
- this symbol "~" is meanig about , in the same order of magnitude; clearly in line 136 or 137 you meant ≈ , otherwise the two values are the same.7
- if is possible you can add a photo of the experimental setup, how the solutions, dispersions, vials etc looks like, the reader migh be more involved.
Author Response
We thank the helpful comments, and suggestions of the reviewer. We have changed the manuscript according to his/her suggestions, point by point. We have highlighted the changes in blue in the new version.
The author present a manuscript regardin the possibility of tuning the morphology in the nanoscale of NH4CN polymers, using MW radiation.
The paper presents all the section completed, the introduction and the conclusion are rich of details and the experimental part is covering well all the aspects. The discussion is well done, all the data presented are commented and overall is well done.
I have only a major concern: on a total of 28 reference 11 cite the corresponding author Ruiz-Bermejo, even if does not exist a maximum I think more than 5 is not correct. Please as in the review in citation 4 you have already cited [21] as 137; [25] as 139; [26] as 140 etc please cite the same review and do not abuse of self-citations.
In agreement with the reviewer in the revised version of our manuscript, we have use the reference 4 and we have deleted the self-citations 21, 24-27. In addition, a new reference has been added, and a total of 24 references have been cited.
others minor comments:
- this symbol "~" is meanig about , in the same order of magnitude; clearly in line 136 or 137 you meant ≈ , otherwise the two values are the same.7
We have change the symbol in accordance with the reviewer´s comment.
- if is possible you can add a photo of the experimental setup, how the solutions, dispersions, vials etc looks like, the reader migh be more involved.
Following the reviewer’s suggestion, in the section “2.1. Synthesis of the NH4CN polymers” a new figure (figure 1, page 3) has been added to illustrate the synthetic process.

Round 2
Reviewer 1 Report
Cristina Pérez-Fernández et al. report Tuning the morphology in the nanoscale of NH4CN polymers synthesized by microwave radiation: A comparative study. I have highly satisfied with the author's response. The additional experiments have improved the quality of the MS. The controlled experiments have opened several windows in NH4CN polymerization.
Thus, it has met the journal's requirements.
Reviewer 2 Report
I thank the authors for having understand my comments and suggestions.